



# Contribution of air-mass transport via the South Asia High to
# the deep stratosphere in summer
Yu Liu[1,*]
[1] State Key Laboratory of Severe Weather, Key Laboratory of Atmospheric Chemistry
of CMA, Chinese Academy of Meteorological Sciences, Beijing 100081, China
[*]corresponding author: yuliu@cma.gov.cn



**Abstract:**  This study proposes a method for estimating meridional and vertical air-mass
transport in the stratosphere based on the mass conservation equation. The method does not
require calculation of the velocity and flux, and avoids the uncertainty in vertical velocity
estimates. Using satellite observations of hydrogen cyanide (HCN) concentrations in summer
(June–September), the relative contributions of air mass originating from the troposphere and
transporting into the deep stratosphere via the tropical tropopause and South Asia High (SAH)
were estimated as 7.72% and 14.55%, while those of HCN were 7.17% and 15.72%,
respectively. These results indicate that the air-mass contributions of the SAH are greater than
those of the tropical troposphere, and that the SAH is the most important air-mass transport
pathway from the troposphere to the deep stratosphere in summer. This suggests that the
impact of pollutants from Asia on the stratosphere is greater than that reported by previous
studies.

**Keywords**: South Asia High, deep stratosphere, air-mass, pathway, contribution




















**Introduction**

The South Asia High (SAH) is a dominant anticyclone of the upper troposphere and lower stratosphere (UTLS) that occurs in summer. Also known as the Asian summer monsoon anticyclone, it covers large areas of Asia, Africa and Europe. It is an important pathway for air-mass transport from the troposphere to the stratosphere, through which pollutants in the boundary layer can quickly enter the stratosphere (Cong et al., 2002; Gettelman et al., 2004; Li et al., 2005; Fu et al., 2006; Randel & Park, 2006; Randel et al., 2010). Transport of pollutants from the boundary layer to the SAH by convective activities is responsible for the peak pollutant concentrations observed in the SAH (Randel & Park, 2006; Park et al., 2007) and typically lasts from June to September (Liu et al., 2003; Randel et al., 2010). Sequentially, the air mass of the SAH can enter the deep stratosphere (Randel et al., 2010; Garny & Randel, 2016), which is the region of stratosphere above the tropical tropopause layer (TTL).

The TTL ranges from 150 hPa to 70 hPa, or from 14 km to 20 km above sea level (Bergman et al., 2012). Air masses originating from the troposphere need to penetrate the TTL before they enter the rising branch of the Brewer-Dobson circulation (BDC), and eventually take part in mass circulation in the deep stratosphere. The effects of different vertical velocities on transport through the TTL have been investigated by trajectory models. Some of the major features of this transport system, such as its time scale, route and diffusion, have been identified to depend primarily on vertical velocity (Ploeger et al., 2010). It was also found that differences in the transport of boundary layer air into the SAH estimated using different reanalysis datasets are mainly due to differences in vertical velocity (Bergman et al., 2013). In general, two methods are usually employed to estimate vertical velocity in the UTLS: the diabatic method and the kinematic method. In the diabatic method, vertical velocity is calculated as the motion across the isentropic surface based on using a diabatic heating rate in an isentropic coordinate system. In the kinematic method, the vertical velocity is estimated by solving the continuity equation in the vertical pressure coordinate. Under conditions of no phase change, the diabatic method is often a better choice than the kinematic method because of its smaller deviations (Wohltmann & Rex, 2008; Ploeger et al., 2010). In addition, vertical velocity can also be obtained from the ascending speeds of tracers tracked by satellites (Mote et al., 1998; Niwano et al., 2003).





It has been shown that, based on differences in vertical velocities, 31% (based on
reanalysis datasets) or 48% (based on the diabatic heating rate) of the SAH air mass on the
360 K isentropic surface can reach the stratosphere if the trajectories are used as an indicator
(Garny & Randel, 2016). This highlights the importance of vertical velocity in quantifying the
air-mass portion that is transported from the SAH into the stratosphere. However, this task
often proves difficult due to uncertainties in vertical velocity estimation. There is evidence
that water vapor transported from the SAH to the stratosphere in summer accounts for about
75% of the total water flux from the troposphere to the stratosphere (Gettelman et al., 2004).
It is suggested that about 20% of the air in the tropical lower stratosphere originates from
atmospheric boundary layers in Asian areas (Orbe et al., 2015). Since all of these results are
based on models or reanalysis datasets, large uncertainties exist. Therefore, further
investigation of the amount of air mass that is transported through the SAH to the deep
stratosphere is warranted.
In this work, we propose a new method for estimating the amount of air-mass transport
through the SAH into the deep stratosphere. The advantage of this method is that it does not
require the calculation of vertical velocities, and uses wind field data. The contribution of
air-mass transport through the SAH into the deep stratosphere is calculated by
satellite-observed tracer data. This method avoids the uncertainty in basing estimates on
vertical velocities.

**Methods and Datasets**
The mass conservation equation is:
$$\frac{\partial C}{\partial t} + u\frac{\partial C}{\partial x} + v\frac{\partial C}{\partial y} + w\frac{\partial C}{\partial z} = Q \qquad (1)$$
where $C$ is the tracer concentration, and $u$, $v$, and $w$ are wind speeds in the zonal ($x$),
meridional ($y$), and vertical ($z$) directions, respectively. Q represents the tracer production or
depletion. In general, when transport in the meridional plane is investigated, the concepts of
mean flow and eddies are applied. However, the partitioning between mean flow and eddies
depends critically on the type of averaging processes used, such as conventional Eulerian
zonal mean, transform Eulerian mean, and generalized Lagrangian mean (Andrews et al.,
1987). These methods calculate the mean flow and parameterize the eddies to investigate





tracer distribution and variation. However, these methods have difficulty in clearly
distinguishing between meridional and vertical transport. In order to avoid the problem, Eq.
(1) is directly averaged in the zonal direction, which becomes:
$$\frac{\partial \bar{c}}{\partial t} + \overline{v\frac{\partial c}{\partial y}} + \overline{w\frac{\partial c}{\partial z}} = \bar{Q} \tag{2}$$

If the wind field is constant in the zonal direction, the velocity components can be outside
the average sign. In a similar way, equivalent wind speeds are defined as follows:
$$v_e(c) = \overline{v\frac{\partial c}{\partial y}} \Big/ \frac{\partial \bar{c}}{\partial y}$$

$$w_e(c) = \overline{w\frac{\partial c}{\partial z}} \Big/ \frac{\partial \bar{c}}{\partial z}$$

Equation (2) becomes:
$$\frac{\partial \bar{c}}{\partial t} + v_e(c)\frac{\partial \bar{c}}{\partial y} + w_e(c)\frac{\partial \bar{c}}{\partial z} = \bar{Q} \tag{3}$$

where $v_e$ (c) and $w_e$ (c) depend on the distribution of trace constituents. Generally, the
transform Eulerian mean transport equation (Andrews et al., 1987) is:
$$\frac{\partial \bar{c}}{\partial t} + \overline{v_t}\frac{\partial \bar{c}}{\partial y} + \overline{w_t}\frac{\partial \bar{c}}{\partial z} = \overline{Q_t} + \rho^{-1}\nabla \cdot \left(\rho \vec{K} \cdot \nabla \bar{c}\right) \tag{4}$$

$$\vec{K} = \begin{bmatrix} k_{yy} & k_{yz} \\ k_{yz} & k_{zz} \end{bmatrix}$$

where $\vec{K}$ is a "diffusion tensor", $(\overline{v_t}, \overline{w_t})$ is the effective transport velocity (Plumb and
Mahlman, 1987; Andrews et al., 1987), and $\rho$ is the air density. Comparing Eq. (4) with Eq.
(3), it is found that:
$$v_e(c) = \overline{v_t} - \rho^{-1}\left(\frac{\partial}{\partial y}\left(\rho\left(k_{yy}\frac{\partial \bar{c}}{\partial y} + k_{yz}\frac{\partial \bar{c}}{\partial z}\right)\right)\right)\Big/\frac{\partial \bar{c}}{\partial y}$$

$$w_e(c) = \overline{w_t} - \rho^{-1}\left(\frac{\partial}{\partial z}\left(\rho\left(k_{zz}\frac{\partial \bar{c}}{\partial z} + k_{yz}\frac{\partial \bar{c}}{\partial y}\right)\right)\right)\Big/\frac{\partial \bar{c}}{\partial z}$$

Therefore, Eq. (3) is correct.
When averaged over a few weeks, the tendency term becomes small (Andrews et al.,
1987). In this work, the averaging period is from June to September, so Eq. (3) is simplified
as:





$$v_e \frac{\partial \overline{c}}{\partial y} + w_e \frac{\partial \overline{c}}{\partial z} = \overline{Q} \tag{5}$$
In the tropical lower stratosphere, the meridional and vertical transports are both inputs
of a tracer originating from the troposphere, such as hydrogen cyanide (HCN). Equation (5)
can be transformed into a discrete form:
$$a \times \left(\overline{C_{k+1,J}} - \overline{C_{k,J}}\right) + b \times \left(\overline{C_{k,J}} - \overline{C_{k,J-1}}\right) = \overline{Q_{k,j}} \tag{6}$$
$$a = \frac{v_e}{\Delta y}$$
$$b = \frac{w_e}{\Delta z}$$
where k and j are the grid indices in the y and z directions, respectively; and a and b are
coefficients.
In the tropical UTLS, it is necessary to add a negative sign before coefficient *a*, because
$v_e$ is the northerly wind. We obtain Eq. (7) by re-arranging Eq. (6):
$$\overline{C_{k,J}} = \frac{a}{a+b}\overline{C_{k+1,J}} + \frac{b}{a+b}\overline{C_{k,J-1}} + \frac{\overline{Q_{i,j}}}{a+b} \tag{7}$$
The first two terms on the right-hand side of Eq. (7) are the contributions to tracer
concentration, $\overline{C_{k,J}}$, from the meridional and vertical transports, respectively. The third term
represents chemical production or depletion.
Hydrogen cyanide (HCN) has a chemical life cycle of about four years (Park et al.,
2013). Therefore, when HCN is used as the tracer, its Q is low in the low stratosphere, such
that the third term can be omitted. Equation (7) then becomes:
$$\overline{C_{k,J}} = \frac{a}{a+b}\overline{C_{k+1,J}} + \frac{b}{a+b}\overline{C_{k,J-1}} \tag{8}$$
We now define $F = \frac{a}{a+b}$; thus, $\frac{b}{a+b} = 1 - F$. Equation (9) can then be derived:
$$F = \frac{\overline{C_{k,j}} - \overline{C_{k,J-1}}}{\overline{C_{k+1,J}} - \overline{C_{k,J-1}}} \tag{9}$$
Analyses of the relationships between tracer concentration terms indicate that F represents
the ratio of meridional air-mass transport, and $1 - F$ represents the proportion of vertical
air-mass transport. The detailed analyses are as follows. Equation (8) becomes:
$$\overline{C_{k,J}} = F\overline{C_{k+1,J}} + (1-F)\overline{C_{k,J-1}} \tag{10}$$
The first term ($F\,\overline{c_{k+1,J}}$) on the right-hand side is the contribution of meridional transport
to the tracer concentration ($\overline{c_{k,j}}$). $F\,\overline{c_{k+1,j}}/\overline{c_{k,j}}$ is the contribution ratio of meridional transport
to the tracer concentration. $F\,\overline{c_{k+1,j}}/\overline{c_{k,j}} \times (\overline{c_{k,j}}\,\overline{\rho_{k,j}})$ is the contribution of meridional



transport to the tracer mass concentration $\left( \overline{c_{k,j}} \, \overline{\rho_{k,j}} \right)$. $\overline{\rho_{k,j}}$ is air density.
$F \, \overline{c_{k+1,j}} / \overline{c_{k,j}} \times \left( \overline{c_{k,j}} \, \overline{\rho_{k,j}} \right) / \overline{c_{k+1,j}}$ is the contribution of meridional transport to the air mass.
$F \, \overline{c_{k+1,j}} / \overline{c_{k,j}} \times \left( \overline{c_{k,j}} \, \overline{\rho_{k,j}} \right) / \overline{c_{k+1,j}} / \overline{\rho_{k,j}}$ is the contribution ratio of meridional transport to the
air-mass, which is equal to F. Hence, F represents the contribution ratio of meridional
transport to the air mass. Similarly, $1 - F$ is the contribution ratio of vertical transport to the
air mass.

Based on Equation (9), the contribution ratio of the air-mass meridional transport from the

northern hemisphere subtropics (NHS, 20°–40°N) to the tropical lower stratosphere can be
obtained.

In the NHS low stratosphere where vertical transport is the only input, Eq. (5) is

transformed into another discrete form, and Q is omitted:
$$-d \times \left( \overline{C_{k+1,J}} - \overline{C_{k,J}} \right) + e \times \left( \overline{C_{k+1,J}} - \overline{C_{k+1,J-1}} \right) = 0 \qquad (11)$$
$$d = \frac{v_e}{\Delta y}$$
$$e = \frac{w_e}{\Delta z}$$
where d and e are coefficients. Rearranging Eq. (11) easily yields Eq. (12):
$$\overline{C_{k+1,J}} = \frac{e}{e-d} \overline{C_{k+1,J-1}} - \frac{d}{e-d} \overline{C_{k,J}} \qquad (12)$$

The first term on the right-hand side represents the input, while the second term is the

output. After defining $G = \frac{e}{e-d}$, thus, $\frac{d}{e-d} = G - 1$, and Eq. (13) follows:
$$G = \frac{\overline{C_{k+1,J}} - \overline{C_{k,J}}}{\overline{C_{k+1,J-1}} - \overline{C_{k,J}}} \qquad (13)$$
G should be greater than 1. Therefore, $\overline{C_{k+1,J}}$ must be higher than $\overline{C_{k+1,J-1}}$. This property
can explain the formations of two high centers of HCN in the SAH and the rising branch of
the Brewer-Dobson circulation (see Fig. 1). It also provides verification of this method.

In this study, HCN was used as a tracer. HCN is produced by the burning of terrestrial

biomass, but is removed over the tropical oceans because of its solubility in water. Its
chemical lifecycle is about four years (Park et al., 2013). HCN often serves as a tracer of
tropospheric pollutants that enter the stratosphere through the SAH (Randel et al., 2010; Park
et al., 2013). The data analyzed in this study were obtained from the ACE-FTS satellite
product (Bernath et al., 2005) and were processed following the method of Park et al. (2013),
then interpolated to a 5°(latitude) × 10°(longitude) grid. The tropopause height in Fig. 1 and





winds in Fig. 2 are ERA-interim reanalysis data (2.5 ° × 2.5 °) from the European Centre for
Medium-Range Weather Forecasts (ECMWF).

**Results**
Figure 1 depicts the mean vertical distribution of HCN from June to September 2004–2010.
Clearly, the tropical troposphere is an area of low HCN concentration, while high
concentrations of HCN occur in the NHS. These high HCN concentrations in the NHS can be
transported into the deep stratosphere by entering the rising branch of the BDC and merge
into the mass circulation in the deep stratosphere. At the same time, in the lower stratosphere
(16–20 km), HCN is transported meridionally from the NHS to the Southern Hemisphere and
descends in the region south of 20 ° S. This is consistent with the results of Randel et al.
(2010). In addition, there are two centers of high HCN: one at 14–16 km in the SAH and
another at 21–25 km in the tropics.
The solid white line in Fig. 1 is the thermal tropopause height calculated from
ERA-interim reanalysis data. It can be seen that the tropopause height in the 20 ° N–40 ° N
region is between 15–16 km, but is close to 16 km in the tropics (20 ° S–20 ° N). It is well
known that an upwelling motion prevails in the tropical stratosphere. Figure 1 demonstrates
that high HCN concentrations are transported from the NHS (20 ° N–40 ° N) to the tropics.
Therefore, the air mass in the tropical lower stratosphere has two inputs: upward transport
from the troposphere and meridional transport from the NHS. High vertical resolution in the
ACE-FTS data allows a detailed estimate of the contribution from the NHS at multiple levels
in the TTL. Based on Eq. (9), the coefficient F is calculated at each level in the TTL. A budget
analysis of the air mass and HCN at 14.5–17.5 km in the tropics is listed in Table 1, including
the mean HCN concentrations in the NHS and tropics, coefficient F and 1 − F, and the
aggregated air-mass and HCN contributions from the NHS at 14.5 km to each level. In the
upper troposphere, the air-mass contributions of the NHS at 14.5 km and 15.5 km are 3.8%
and 7.14%, respectively. Evidently, the greatest air-mass contributions from the NHS occur at
16.5 km and 17.5 km, accounting for 22.25% and 26.98%, respectively.
As can be seen from Table 1, the contribution of meridional transport from the NHS to
the air mass increases with height: from 3.8% at 14.5 km to 26.98% at 17.5 km. Meanwhile,





the contribution of vertical transport to the air mass gradually decreases from 96.2% at 14.5
km to 73.02% at 17.5 km. Although the contribution of the vertical transport is about
three-fold greater than that of meridional transport at 17.5 km, the aggregated contribution of
meridional transportation from the NHS to the air mass is 49.25% at 17.5 km, which is close
to the contribution of vertical transport from the tropical troposphere. Like its contribution to
the air mass, the contribution of meridional transport to HCN gradually increased from 4.74%
at 14.5 km to 29.77% at 17.5 km, while the aggregate contribution to HCN reached 55.64% at
17.5 km, thereby exceeding the contribution of vertical transport from the tropical
troposphere.

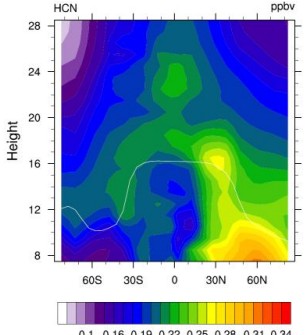

Figure 1. Time- and zonal-averaged mixing ratio (ppbv) of HCN in summer (June to
September) derived from ACE-FTS satellite measurements. The white line denotes the
tropopause height.
**Table 1.** Contributions of meridional transport from the NHS1 to air masses and HCN
concentrations in the tropics at heights of 18.5 km to 20.5 km

| Height (km) | NHS HCN concentration (ppbv) | Tropical HCN concentration (ppbv) | F (%) | (1 − F) (%) | Aggregate air mass from NHS (%) | Aggregate HCN from NHS (%) |
|---|---|---|---|---|---|---|
| 13.5 | 0.2417 | 0.1946 |  |  |  |  |
| 14.5 | 0.2449 | 0.1965 | 3.80 | 96.2 | 3.80 | 4.74 |
| 15.5 | 0.2559 | 0.2007 | 7.14 | 92.86 | 10.67 | 13.41 |
| 16.5 | 0.2601 | 0.2139 | 22.25 | 77.75 | 30.55 | 36.84 |
| 17.5 | 0.2454 | 0.2224 | 26.98 | 73.02 | 49.29 | 55.64 |

Notes: Coefficient F represents the contribution of meridional transport from the NHS. Coefficient 1
− F represents the contribution of vertical transport from below. Column 6 shows the aggregate
contributions of air mass from the NHS at 14.5 km to each layer. Column 7 shows the aggregate
contributions of HCN from the NHS at 14.5 km to each layer.



Figure 2 shows the horizontal distributions of HCN concentrations and wind fields at six
heights from 16.5 km to 21.5 km. It can be seen from their changes with the height that the
influence of the SAH gradually decreases with the height up to 20.5 km. Figure 1 shows that
from 18.5 km to 20.5 km, the tropical upward vertical transport channel (at $10\,^\circ$S to $10\,^\circ$N)
narrows. Moreover, the budget analysis also illustrates that the tropical upward channel
becomes narrow. Table 2 lists the mean HCN concentrations, coefficient F, and $1 - F$ in the
tropics ($10\,^\circ$S–10 $^\circ$N) from 18.5 km to 20.5 km, as well as the aggregate contribution of
meridional transport from the subtropical Northern Hemisphere (NHS1, $10\,^\circ$N–$40\,^\circ$N) to air
mass and HCN concentrations. Table 2 shows that the contributions of meridional transport to
air mass are 16.67% at 18.5 km, 69.81% at 19.5 km, and 39.47% at 20.5 km. The aggregate
contributions of meridional transport to air mass and HCN at 20.5 km are 92.28% and 92.83%,
respectively. So, only 7.72% of the air mass and 7.17% of the HCN concentration come from
the tropical troposphere.
In order to determine the contribution of meridional transport originating from the SAH
to the tropical air mass and HCN, air mass and HCN inputs and outputs in the NHS from 14.5
km to 17.5 km were analyzed (Table 3). Table 3 shows that the HCN concentrations gradually
increased from 0.2417 ppbv at 13.5 km to 0.2601 ppbv at 16.5 km. The distributions of HCN
from 14.5 km to 16.5 km are in accordance with Eq. (12). According to Eq. (13), coefficient
G is 107.08% at 14.5 km, 124.89% at 15.5 km, and 110.0% at 16.5 km. These results indicate
that the air mass and HCN inputs to the NHS all originate from vertical transport.
The horizontal distributions of HCN at 16.5 km height (Fig. 2) show that a high center of
HCN largely overlaps the SAH ($20\,^\circ$N–$40\,^\circ$N, $30\,^\circ$E–$120\,^\circ$E) and high HCN concentrations
are transported to surrounding areas as far as the Southern Hemisphere via the tropics. The
horizontal wind fields were obtained from ERA-interim reanalysis data. From 14.5 km to 16.5
km, the HCN concentrations in the NHS are higher than in both the tropics and the
mid-to-high latitudes of the Northern Hemisphere. The inputs of these levels in the NHS are
entirely from upward vertical transport. As mentioned above, the SAH is the primary upward
vertical transport pathway in the NHS. It is plausible that the HCN in the NHS originates
from the SAH before being further transported to the tropics (Randel et al., 2010).



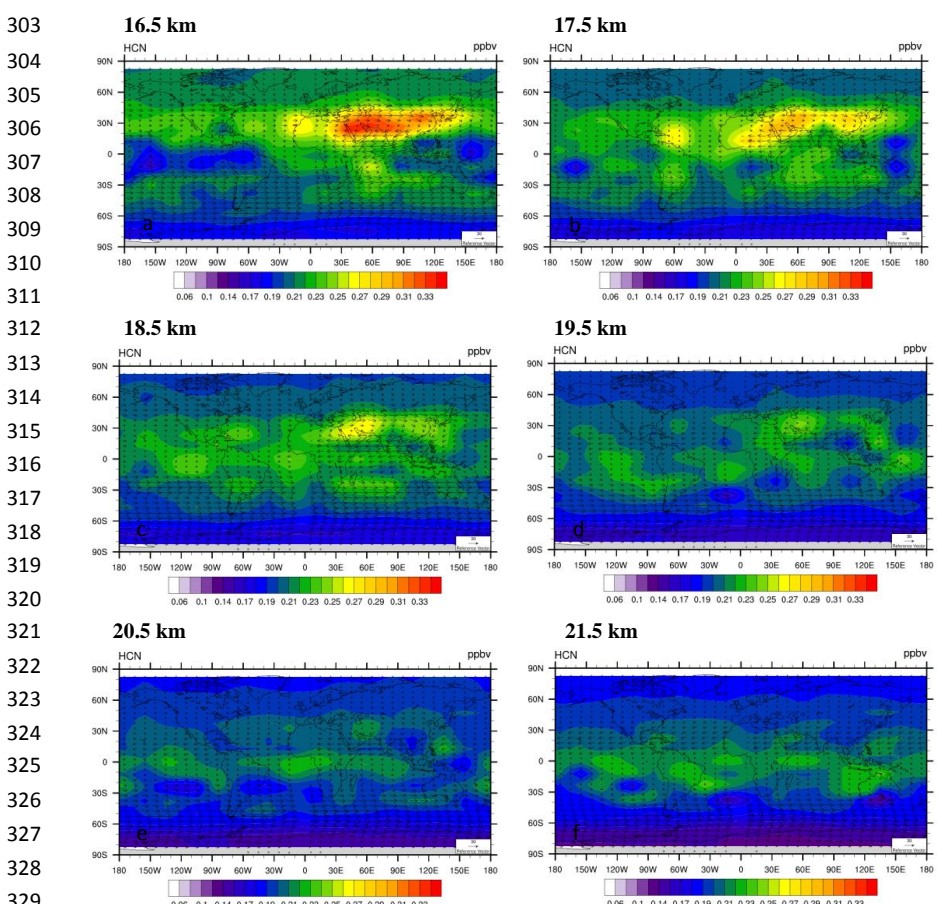

Figure 2. 2004–2010 averaged mixing ratio (ppbv)] of HCN at heights of 16.5 km (a), 17.5 km (b), 18.5 km (c), 19.5 km (d), 20.5 km (e), and 21.5 km (f) in summer (June-September), as derived from ACE-FTS observations.

At 17.5 km, the HCN concentration in the NHS is 0.2454 ppbv, which is lower than that at 16.5 km (0.2601 ppbv). The distribution of HCN concentrations is in accordance with Eq. (8). HCN has two inputs: vertical transport, and meridional transport from the mid-latitudes of the Northern Hemisphere ($40\,°N$–$60\,°N$), where the concentration of HCN is 0.2173 ppbv. According to Eq. (9), coefficient F is 34.35% and $1 - F$ is 65.65%. This illustrates that the contributions of vertical transport to the air mass and HCN were 65.65% and 69.58%, respectively. These are also the contributions originating from the SAH.

From 18.5 km to 20.5 km, because the tropical upward vertical transport channel occurs from $10\,°S$ to $10\,°N$, the Northern Hemisphere subtropical area was also adjusted to $10\,°N$–$40\,°$



N. The mid-latitudes of the northern hemisphere are still between $40\,°\text{N}–60\,°\text{N}$. A budget
analysis of the air mass and HCN concentrations in the Northern Hemisphere subtropics is
shown in Table 4. Like the situation at 17.5 km, based on Eq. (9), the coefficient F is 49.09%
at 18.5 km, 61.57% at 19.5 km, and 60.29% at 20.5 km. The contributions to air mass
originating from the SAH are 33.42% at 18.5 km, 12.84% at 19.5 km, and 5.10% at 20.5 km.
Similarly, the HCN contributions originating from the SAH are 35.42% at 18.5 km, 13.61% at
19.5 km, and 5.40% at 20.5 km.

**Table 2.** Contributions of meridional transport from the NHS1 to air masses and HCN
concentrations in the tropics at heights of 18.5 km to 20.5 km


| Height (km) | NHS1 mean HCN (ppbv) ($10\,°–40\,°\text{N}$) | Tropical mean HCN (ppbv) ($10\,°\text{S}-10\,°\text{N}$) | F (%) | (1−F) (%) | Aggregate air mass from NHS (%) | Aggregate HCN from NHS (%) |
|---|---|---|---|---|---|---|
| 17.5 | 0.2415 | 0.2211 | | | | |
| 18.5 | 0.2253 | 0.2218 | 16.67 | 83.33 | 57.74 | 63.15 |
| 19.5 | 0.2112 | 0.2144 | 69.81 | 30.19 | 87.24 | 88.50 |
| 20.5 | 0.2030 | 0.2099 | 39.47 | 60.53 | 92.28 | 92.83 |

Notes: Coefficient F represents the contribution of meridional transport from the NHS1. Coefficient 1 −
F represents the contribution of vertical transport from below. Column 6 shows the aggregate
contributions of air mass from the NHS1 at 14.5 km to each layer. Column 7 shows the aggregate
contributions of HCN from the NHS1 at 14.5 km to each layer.

**Table 3.** Contributions of vertical transport from the SAH to air masses and HCN
concentrations in the NHS at heights of 14.5 km to 17.5 km


| Height (km) | NHS HCN (ppbv) | Tropical HCN in (ppbv) | G (%) | (G − 1) (%) | HCN in NML | F (%) | 1 − F (%) | Contribution of HCN from SAH |
|---|---|---|---|---|---|---|---|---|
| 13.5 | 0.2417 | 0.1946 | | | | | | |
| 14.5 | 0.2449 | 0.1965 | 107.08 | 7.08 | | | | |
| 15.5 | 0.2559 | 0.2007 | 124.89 | 24.89 | | | | |
| 16.5 | 0.2601 | 0.2139 | 110.0 | 10.0 | | | | |
| 17.5 | 0.2454 | 0.2224 | | | 0.2173 | 34.35 | 65.65 | 69.58 |

Notes: Coefficient G represents the contribution of vertical transport from the SAH. G − 1 represents
the output of meridional transport. Column 6 shows mean HCN concentrations in mid-latitudes of
the Northern Hemisphere (NML). Coefficient F represents the contribution of meridional transport
from the NML. Coefficient 1 − F represents the contribution of vertical transport from the SAH.



Column 9 shows the contribution of HCN from vertical transport from the SAH.

The contributions of the SAH to tropical air mass and HCN concentrations can be further
analyzed after obtaining 1) the contributions of tropical vertical and meridional transport and
2) the SAH's contributions to air mass and HCN in the Northern Hemisphere subtropics.
Table 5 shows the contributions originating from the SAH to the tropical air mass and HCN.
The second column shows the contribution of meridional transport from the NHS to the
tropical air mass, with the maximum of 69.81% occurring at 19.5 km. The third column
shows the contributions of vertical transport from the SAH to the NHS air mass, which are
100% at 14.5–16.5 km, then gradually decrease from 65.65% at 17.5 km to 5.10% at 20.5 km.
The fourth column shows the contributions of the SAH to the tropical air mass via meridional
transport, with the maximum of 22.25% occurring at 16.5 km and the minimum of 2.1%
occurring at 20.5 km. The fifth column shows the aggregate contributions of SAH air from
14.5 km to the current height to the tropical air mass, which peaks at 40.02% at 17.5 km. The
aggregate contribution gradually decreases between heights of 17.5 km and 20.5 km, where it
reaches 14.55%. The sixth column shows the aggregate contribution of HCN from the SAH to
the tropics, which peaks at 45.40% at 17.5 km. Similar to the aggregate contributions to air
mass, the aggregate contribution of HCN gradually decreases between 17.5 km and 20.5 km,
where it is 15.72%. The seventh column shows the air mass contribution of the tropical
troposphere, which constantly decreases with height and reaches 7.72% at 20.5 km. Similarly,
the eighth column shows the contribution of HCN from the tropical troposphere, which
gradually decreases with height and is 7.17% at 20.5 km.
Figure 1 shows a HCN high center in the tropics from 21.5 km to 25.5 km. The HCN
distributions are in accordance with Eq. (12) at 21.5 km and 22.5 km, whose coefficient G
values are 257.89% and 235.08% based on Eq. (13), respectively. This illustrates that the air
mass and HCN inputs all come from vertical transport. The tropical upward channel links the
rising branch of the Brewer-Dobson circulation. As can be seen from Table 5, the contribution
of the SAH to the air mass of the rising branch of BDC is 14.55%, and the contribution
originating from the tropical troposphere is 7.72%. Therefore, the contribution of the SAH is
1.88 times that of the tropical troposphere. The contributions of the SAH and tropical





troposphere to HCN concentrations are 15.72%, and 7.17%, respectively. The contribution of
the SAH to HCN concentration is 2.19 times that of the tropical troposphere. This indicates
that the amounts of air mass and HCN originating from the troposphere and transporting
through the SAH into the deep stratosphere in summer exceeds those coming from the
tropical troposphere. Therefore, the SAH is the most important pathway of air-mass transport
from the troposphere to the deep stratosphere in summer.

**Table 4.** Contributions of vertical transport from the SAH to air masses and HCN
concentrations in the NHS1 at heights of 18.5 km to 20.5 km


| Height (km) | HCN in NHS1 (ppbv) (10°–40°N) | HCN in NML (ppbv) (40°–60°N) | F (%) | 1 − F (%) | Contribution of air mass from SAH | Contribution of HCN from SAH |
|---|---|---|---|---|---|---|
| 17.5 | 0.2415 | 0.2173 | | | | |
| 18.5 | 0.2253 | 0.2085 | 49.09 | 50.91 | 33.42 | 35.42 |
| 19.5 | 0.2112 | 0.2024 | 61.57 | 38.42 | 12.84 | 13.61 |
| 20.5 | 0.2030 | 0.1976 | 60.29 | 39.71 | 5.10 | 5.40 |


**Table 5.** Contributions of transport from the SAH to air masses and HCN
concentrations in the tropics


| Height (km) | Air mass contribution from NHS to tropics (%) | Vertical transport of air mass from SAH to NHS (%) | Meridional transport of air mass from SAH to tropics (%) | Aggregate air mass from SAH to tropics (%) | Aggregate HCN from SAH to tropics (%) | Contribution of vertical transport of air mass to tropics | Contribution of air mass from tropical troposphere to tropics | HCN contribution from tropical troposphere to tropics |
|---|---|---|---|---|---|---|---|---|
| 13.5 | | | | | | | | |
| 14.5 | 3.80 | 100.0 | 3.80 | 3.80 | 4.74 | 96.2 | 96.2 | 95.26 |
| 15.5 | 7.14 | 100.0 | 7.14 | 10.67 | 13.41 | 92.86 | 89.33 | 86.59 |
| 16.5 | 22.25 | 100.0 | 22.25 | 30.55 | 36.84 | 77.75 | 69.45 | 63.16 |
| 17.5 | 26.98 | 65.65 | 17.71 | 40.02 | 45.40 | 73.02 | 50.71 | 44.36 |
| 18.5 | 16.67 | 33.42 | 5.57 | 38.92 | 43.35 | 83.33 | 42.26 | 36.85 |
| 19.5 | 69.81 | 12.84 | 8.96 | 20.71 | 22.35 | 30.19 | 12.76 | 11.50 |
| 20.5 | 39.47 | 5.10 | 2.01 | 14.55 | 15.72 | 60.53 | 7.72 | 7.17 |




**Discussion and Conclusion**

The results show that there are two main transport channels from the troposphere to the stratosphere in summer: one in the tropics and one in the SAH. Air originating from the troposphere needs to pass through the TTL before it can enter the deep stratosphere and take part in mass circulation there. The contribution of air originating from the tropical troposphere to the air mass of the rising branch of the BDC is 7.72%, and that coming from the SAH is 14.55%. Hence, the contribution of air from the SAH is 1.88 times that originating from the tropics. This is somewhat different to conventional knowledge. The contribution of air originating from the tropical troposphere has been gradually reduced, although vertical transport takes the main role in the TTL except at 19.5 km; however, meridional transport continuously dilutes the air originating from the tropical troposphere. The maximum contribution of air from the SAH to the tropical air mass is 22.25% at 16.5 km, and the maximum aggregate contribution is 40.02% at 17.5 km. The aggregate contribution of air from the SAH gradually decreases from 18.5 km to 20.5 km, but some air from the SAH is contributed via meridional transport, so the decrease in air contributed by the SAH is more gradual than that originating from the tropical troposphere. Specifically, the aggregate contributions from the SAH at heights of 14.5–18.5 km to the deep stratosphere are 7.11% for air mass and 8.43% for HCN, while the contributions of the SAH at heights of 19.5–20.5 km are 7.44% for air mass and 7.29% for HCN. Therefore, the contributions from the two layers are comparable. The aggregate contribution of the air from the SAH to the air mass of the rising branch of the DBC finally exceeds that of the tropical troposphere. Hence, the SAH is the most important pathway of air-mass transport from the troposphere to the deep stratosphere in summer. This suggests that the impact of pollutants from Asia on the stratosphere is greater than that reported by previous researches.

The present results contrast with those of Ploeger et al. (2017), who showed that air mass from the SAH hardly enters the deep stratosphere at all in summer. The difference between these results is due to differences in the observed HCN data used by the present study and the estimates of the CLaMS model used by Ploeger et al. (2017). Transports to the Southern Hemisphere and deep stratosphere based on observed HCN data are greater than those based on modelled estimates. The present study reports stronger transport to the deep stratosphere,





and that about 14% of the summer deep stratosphere air mass comes from the SAH.

The contribution of air mass to the deep stratosphere via the SAH also can be estimated

using wind field data to calculate the flux, or by the trajectory method; however, there are
some uncertainties in these methods due to error in vertical velocity estimates. My new
method does not require the calculation of vertical velocity and can estimate the contributions
of meridional and vertical transport based on the distribution of trace components. However,
it cannot derive speed and flux values. The new method explains the formation of a high HCN
center in the SAH and DBC, which also indicates that it is credible.

There are two sources of error in my method: error due to approximation and error in

HCN concentration data. Approximation error involves two factors; one is caused by
overlooking the time tendency term. The time tendency term becomes small when averaged
over a few weeks (Andrews et al., 1987) . Work of Rendel et al.(2010) depicted HCN
latitude-time variation from 2004 to 2009 for low stratosphere (16-23 km) (Figure 3, Rendel
et al., 2010). The results shown that HCN variation from June to September is relatively small
and air-mass from the $40^o\,N$ along meridional direction is quickly transported to $20^o\,S$. This
also illustrate that HCN tendency term from June to September is much less than the
meridional transport term. Because my study period was four months, this error is small. The
second factor is caused by omitting the production and depletion term; however, because the
chemical lifetime of HCN is four years (Park et al., 2013), this error should be very small. If
production and depletion are considered, then because HCN only undergoes depletion, the
estimated contribution of air from the SAH will be higher. Errors in the satellite observation
of HCN concentrations may also cause errors in my estimates. However, between 14.5 km
and 20.5 km, such errors are less than 10%, suggesting that my estimates should be
reasonable.

**Acknowledgments** This work was supported by the National Natural Science Foundation of
China (91537213, 91837311). The ACE mission is funded primarily by the Canadian Space
Agency. Citation: Bernath et al. (2005) https://doi.org/10.1029/2005GL022386. ACE data:
http://www.ace.uwaterloo.ca/data.php



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





Table Caption List

Table 1. Contributions of meridional transport from the NHS to air masses and HCN
concentrations in the tropics at heights of 14.5 km to 17.5 km

Table 2. Contributions of meridional transport from the NHS1 to air masses and HCN
concentrations in the tropics at heights of 18.5 km to 20.5 km

Table 3. Contributions of vertical transport from the SAH to air masses and HCN
concentrations in the NHS at heights of 14.5 km to 17.5 km

Table 4. Contributions of vertical transport from the SAH to air masses and HCN
concentrations in the NHS1 at heights of 18.5 km to 20.5 km

Table 5. Contributions of transport from the SAH to air masses and HCN
concentrations in the tropics




Figure Caption List

Figure 1. Time and zonal averaged mixing ratio (ppbv) of HCN in summer (June to
September) derived from ACE-FTS satellite measurements. The white line denotes
the tropopause height.

Figure 2. 2004-2010 averaged mixing ratio (ppbv)] of HCN at 16.5 (a), 17.5 (b), 18.5
(c), 19.5 (d), 20.5 (e) and 21.5 km (f) in summer (June to September) derived from
ACE-FTS observations.





