# Peer review of "Contribution of air-mass transport via the South Asia High to"

_Atmospheric Chemistry and Physics, 2020_

## Short Comment (SC1) · 5 Nov 2020

A method to assess transport by application of the continuity equation to measured trace gas fields has been published by von Clarmann and Grabowski (Atmos. Chem. Phys., 16, 14563-14584, 2016, https://acp.copernicus.org/articles/16/14563/2016/). This method has been developed further and applied to trace gas fields by von Clarmann et al. (Atmos. Chem. Phys. Discuss., https://doi.org/10.5194/acp-2019-704, 2019) and was validated by von Clarmann and Grabowski (Atmos. Chem. Phys. Discuss., https://doi.org/10.5194/acp-2020-72, 2020.). To vindicate the claim of novelty of the method by Yu Liu I think it would, at the very least, be adequate to pinpoint where the new method substantially diverges from the one by von Clarmann and Grabowski.

---

## Referee Comment (RC1) · Mohamadou Diallo (Referee) · 8 Dec 2020

General points:

The manuscript presents a method for estimating meridional and vertical air-mass transport from the Asian Summer Monsoon (ASM) into the stratosphere using hydrogen cyanide (HCN) from the ACE-FTS satellite observations. This proposed alternative method to model simulations is interesting and promising, but there are several issues, which need to be clarified. These issues concern the uncertainty in the method hypothesis; uncertainty in the application of the method to ASM (sensitivity to the averaging area; lack of rigorous analysis, the sensitivity of the method to the used HCN trace gas), the uncertainty in the results (error in the observations, coarse latitudes,

and longitudes resolutions), the lack of recent research update regarding the transport pathways and their contributions (a recent article in the SPARC StratoClim project). In addition, the motivation of the paper as claimed by the author that "there are some uncertainties in the model analysis methods due to error in vertical velocity estimates" is not convincing because the numerical approximation uncertainty in the calculation of the vertical and meridional velocities is the same as the numerical uncertainty in this method (e.g. equation 5 for instance) and therefore is very likely in the same order of magnitude in this method. The main uncertainties in the model-based analysis of the SAM pathway contributions are related to the differences in the reanalysis data, the different levels used to initialize the air parcels, and to the different levels at which the estimated contribution of the vertical and horizontal transport of monsoon air is made, therefore, rendering it difficult for inter-comparison values in different studies. Finally, most of the sections and paragraphs in the manuscript are poor, therefore, they need to be revised by enhancing the discussion about the scientific content and the results presentations, and improving the quality of the paper by discussing the appropriate and recent studies where it needed. Based on the amount of work, which needs to be done to improve the manuscript, I recommend very major revisions. It's of particular importance to clarify these major points. In the following here are my major points and specific concerns:

Major points:

1. Regarding the uncertainty in the method hypothesis of neglecting the time tendency term needs to be estimated and its contribution quantified. The uncertainty related to neglecting the production and depletion term of the HCN can be addressed using a second trace gas such SF6, CO, or CO2 or others ACF-TFS or MLS observations (or SWOOSH) to evaluate the sensitivity of the vertical and meridional contributions. This will shed the light on the sensitivity to the used HCN trace gas.

2. One thing very curious is that fraction term G is not closed. Normally, G, as well as F, should be smaller or equal to (or 100%) but G does not hold that distribution

hypothesis.

3. Concerning the uncertainty in the application of the method to ASM area, including the sensitivity to the averaging area; lack of rigorous analysis, sensitivity to the HCN trace gas, the author needs to first to estimate the ASM area at different levels using the PV contours; lapse rate tropopause (LRT) height; LRT potential temperature or cold point tropopause (CPT) height to identify the monsoon area, which contained the ASM air before averaging. In the area where there is not a clearly defined contour, on can tag the concentration of HCN mixing ratios which are above background levels. The author averaging areas of ASM air (20-40N or 10-40N) are very arbitrary chosen therefore induced uncertainties in the estimated values.

4. Other sources of uncertainty in the results concern the error in the observations, particularly the coarse latitudes and longitudes resolutions of the observations. The $5°$ (latitude) $\times 10°$ (longitude) grid can lead to difficulties of defining the ASM area.

5. Most of the paragraphs are poorly written, appropriate references are not properly used at some places, and some sentences are vague (not specific). The updated research publications regarding the transport pathways and their contributions in the SPARC StratoClim project should be addressed. Please find a list of publications below. Minor points:

1. Page 1, line 1, the title can be changed to "Contribution of air-mass transport via the Asian Summer Monsoon anticyclone into the stratosphere" as the ASM is more commonly used and "deep" is more lowe and middle stratosphere than "deep".

2. Page 3, lines 82-83, this paragraph is misleading because the region located above the TTL is not "deep stratosphere" but just stratosphere, which divided into the lower, mid and upper stratosphere. Please rephrase it.

3. Page 4 line 112, this sentence is not correct because please see my general comments.

4. Page 4, line 119, satellite observations have also some uncertainties in the UTLS. The question is how large the uncertainty claimed here in the vertical velocity between different reanalyses compares to the uncertainty in the satellite observations in this region? The differences between kinematic & diabatic are not a result of the calculation but rather a reanalysis of intrinsic issues. In addition, reanalysis-driven simulations compared very well to the in situ observations. Please references below.

5. Page 6 line 177, please replace "ratio" and "portion" with "fraction".

6. Page 6-7, section "Method & Model", please keep the notation of the equation consistent. For instance, we have upper-case and lower-case of the "c".

7. Page 7, line 194-201, the description "G" is not necessary as it's the same as F but in the Asm area.

8. Page 7, line 205-211, this description of the data is very poor. No vertical resolution, uncertainty in the observations, period of the observations, previous studies using the observations are missing. In addition, the ERA-interim tropopause and wind should be both described here instead of all over the place in the manuscript.

9. Page 8, line 231-233, what is the vertical resolution of HCN observations?

10. Page 8, line 239, How these numbers compare with the previous literature?

11. Page 9, line 244-246, is there any explanation of the quasi-equal contribution of the vertical and meridional ASM air?

12. Page 9, line 248-250, is there any explanation of why are there so many differences in the HCN and air mass reported values?

13. Figure 1 caption, please add the period of the observations and make sure that the figure is enough described in the caption. Same argument for figure 2.

14. Page 10, line 280, the selection of averaged area is very arbitrary. Please see my suggestions in the general comments.

15. Page 10, line 291, the values of G look weird as G is a distribution, it should be smaller than 100. This suggests that the method might fail.

16. Page 10, line 299-300, the ASM is an upward motion plus horizontal transport at the outflow level not only a vertical motion. Please rephrase it.

17. Page 15, line 420-424, all this paragraph, in particular the ASM contribution 2 times greater than the tropics need to be proved after addressing the major points.

References:

1) Yan, X., Konopka, P., Ploeger, F., Podglajen, A., Wright, J. S., Müller, R., and Riese, M.: The efficiency of transport into the stratosphere via the Asian and North American summer monsoon circulations, Atmos. Chem. Phys., 19, 15629–15649, https://doi.org/10.5194/acp-19-15629-2019, 2019.

2) Nützel, M., Dameris, M., and Garny, H.: Movement, drivers andbimodality of the South Asian High, Atmos. Chem. Phys.,16, 14755–14774, https://doi.org/10.5194/acp-16-14755-2016,2016.

3) Bucci, S., Legras, B., Sellitto, P., D'Amato, F., Viciani, S., Montori, A., Chiarugi, A., Ravegnani, F., Ulanovsky, A., Cairo, F., and Stroh, F.: Deep-convective influence on the upper troposphere–lower stratosphere composition in the Asian monsoon anticyclone region: 2017 StratoClim campaign results, Atmos. Chem. Phys., 20, 12193–12210, https://doi.org/10.5194/acp-20-12193-2020, 2020.

4) Legras, B. and Bucci, S.: Confinement of air in the Asian monsoon anticyclone and pathways of convective air to the stratosphere during the summer season, Atmos. Chem. Phys., 20, 11045–11064, https://doi.org/10.5194/acp-20-11045-2020, 2020.

5) Vogel, B., Müller, R., Günther, G., Spang, R., Hanumanthu, S., Li,D., Riese, M., and Stiller, G. P.: Lagrangian simulations of the transport of young air masses to the top of the Asian monsoon anticyclone and into the tropical pipe, Atmos. Chem. Phys., 19,6007–6034, https://doi.org/10.5194/acp-19-6007-2019, 2019.

6) Pan, L. L., Honomichl, S. B., Kinnison, D. E., Abalos,M., Randel, W. J., Bergman, J. W., and Bian, J.: Trans-port of chemical tracers from the boundary layer to strato-sphere associated with the dynamics of the Asian summer monsoon, J. Geophys. Res.-Atmos., 121, 14159–14174, https://doi.org/10.1002/2016JD025616, 2016.

7) Vogel, B., Günther, G., Müller, R., Grooß, J.-U., and Riese, M.:Impact of different Asian source regions on the composition of the Asian monsoon anticyclone and of the extratropical lowermost stratosphere, Atmos. Chem. Phys., 15, 13699–13716,https://doi.org/10.5194/acp-15-13699-2015, 2015.

8) Ploeger, F., Günther, G., Konopka, P., Fueglistaler, S., Müller, R.,Hoppe, C., Kunz, A., Spang, R., Grooß, J.-U., and Riese, M.:Horizontal water vapor transport in the lower stratosphere fromsubtropics to high latitudes during boreal summer, J. Geophys.Res., 118, 8111–8127, https://doi.org/10.1002/jgrd.50636, 2013.

9) Ploeger, F., Gottschling, C., Griessbach, S., Grooß, J.-U., Guenther,G., Konopka, P., Müller, R., Riese, M., Stroh, F., Tao, M., Unger-mann, J., Vogel, B., and von Hobe, M.: A potential vorticity-based determination of the transport barrier in the Asian summermonsoon anticyclone, Atmos. Chem. Phys., 15, 13145–13159,https://doi.org/10.5194/acp-15-13145-2015, 2015.

10) Santee, M. L., G. L. Manney,N. J. Livesey, M. J. Schwartz,J. L. Neu, and W. G. Read (2017),A comprehensive overview of the climatological composition of the Asian summer monsoon anticyclone based on 10 years of Aura MicrowaveLimb Sounder measurements,J. Geophys. Res. Atmos.,122, 5491–5514,doi:10.1002/2016JD026408.

11) ACE-FTS observations: http://www.ace.uwaterloo.ca/data.php

---

## Referee Comment (RC2) · Anonymous Referee #2 · 18 Jan 2021

This manuscript is word to word identical to a previous submission in 2019 that I already reviewed. A new read did not induce me to change my report.

This work is based on assuming that the sole consideration of the distribution of a tracer allows to determine quantitatively the transport properties. It relies first on the miss-interpretation of the Eulerian-mean transport equation of Andrews et al., which is rewritten as equation (5) but then the ve and we in this equation are not the same as that defined on lines 143 and 144 and they are not either the effective transport velocity. The basic assumption that transport is just proportional to the mean gradient is usually wrong in the atmosphere and this is precisely what the Eulerian mean theory was aiming at showing. Recasting it as (5) is loosing the point and the paper is implicitly based on this wrong assumption. Then the most serious flaw is in interpreting equation

(10) with F defined in (9) as a transport equation while it is only an arbitrary identity among discretized quantities. This accident occurred due to the double replacement made in equations (6) and (7). Actually F is defined as a/a+b but the discussion is entirely based on (10) and (9). An other arbitrary identity is equation (13) with the additional wrong interpretation that G should be larger than 1 with no reason. There are other curiosities such as changing the sign of the meridional velocity as if it was not counted as positive along the oriented y axis. The "theory" is then applied to the HCN data from ACE, but no results can be considered as valid when the basis is unjustified. Figure 1 is just a redrawing of Figure 2 of Randel et al. (Science, 2010, doi: 10.1126/science.1182274) without mentioning it. Figure 2 a is also very similar to figure 1a of Randel et al. Therefore, the patterns are not surprisingly consistent with this previous work. It is not even discussed why a method which is derived as a zonal mean should apply to the transport by the SAH which is obviously localized in longitude. My recommendation is that the manuscript should be rejected.

---

## Author Comment (AC1) · 1 Mar 2021

Thank Dr. Clarmann for his question. The reply is as follow.

Although your and my methods are both derived from mass conservation equation based on the tracer distributions, but your method focuses on the deviation of wind vector and mixing coefficient to minimize residual between modeled atmospheric state and measurement of atmospheric state; my method is to avoid solving velocities and obtain relative contribution of meridional and vertical transport. Therefore, the both are quite different. I'll introduce your work in the introduction section.

Please also note the supplement to this comment:
https://acp.copernicus.org/preprints/acp-2020-990/acp-2020-990-AC1-supplement.pdf

---

## Author Comment (AC2) · 1 Mar 2021

Thank Dr. Diallo for his comments, questions, guidance and suggestions. The replies to his major and minor questions are as follows:

Replies for the major questions

1. Equation (5) ignores the time tendency term. If the time tendency term remains, it can be treated as a source or sink term. When the tendency is positive, it is a sink; when the tendency is negative, it is a source. Its relative influence is

$-1/(a+b)\ 1/c\ (\partial c)/\partial t$

According to Figure 3 of Rendel et al. (2010), the HCN difference between June and

[Figure]

September is less than 0.02 ppbv, and the HCN average concentration of HCN is more than 0.2 ppbv in the tropical. The air-mass from mid latitude of northern hemisphere can be transported to the tropical within one month, 1 / (a + b) is less than 1 / a, and 1 / a is taken as 1 month; the average duration is 4 months (June to September), so the influence of time tendency term is about less than 2.5 percent.

The chemical process of HCN is depletion role, which is a sink. Its chemical lifetime is about 4 years (Park et al., 2013). The CAM5 model results show that the chemical lifetime of HCN is 2.91 years (about 35 months) at 90 hPa in the tropical. The model top is 1.8 hPa, and vertical levels are 56 layers, the model including troposphere and stratosphere chemistry. Its influence is

1/(a+b) Q/c

which is about 2.9 percent. Compared with the roles of the transportation from meridional and vertical directions, the influences of the time tendency time and sink term are smaller. So, both are ignored.

2. The contribution of the vertical transport to the local is 1 (=100 percent), and takes as reference. G is total transportation that includes the contribution to local and meridional transport, and is equal to e / (e-d) ( > 1). The contribution of the vertical transport to meridional transport is d / (e-d), that is equal to G-1. So, G is closed. The mass is conserved.

3-4. There are several ways to define the scope of ASM. Because of the coarse resolution of data, it is difficult to define the range of ASM accurately. However, the upward transportation of the ASM is the key of the ASM, so it is OK to determine the main area of AMS, which includes the upward transport area of the ASM. Therefore, based on wind fields and HCN distributions in the UTLS I identified the 20-40°N region, which includes the ascending region of the AMS. And it can avoid overestimating the impact of the AMS. The observation error of satellite data is less than 10

Above 18.5 km, the tropical rising area becomes narrower from 20°S-20°N to 10°S-10°N. First, it can be seen from Figure 1. Second, if 20°S-20°N was taken as the rising region, the distribution of HCN does not satisfy the equations (10) and (12). In addition, plodger et al. (2017) selected 10°S-10°N as the tropical rising region. Therefore, the tropical rising region becomes 10°S-10°N above 18.5 km. The ASM area is still 20-40°N. The case of region 10-20°N needs to be considered. Since the rising speed of this region is 0, the meridional variation is 0 in 10- 20°N ( $(\partial c)/\partial y=0$). Therefore, HCN in this region is equal to HCN from 20-40°N. The two areas are put together to treat for easy.

5. According the suggestions, the paper is revised.

Replies for the minor questions

1. According to the suggestion, The title is revised.

2. According to the suggestion, The "deep stratosphere" is revised.

3. According to the suggestion, the relative contents are revised.

4. I think that there are differences between modeled and observed HCN in the southern hemisphere and tropical middle stratosphere in summer (Park et al., 2013; Ploeger et al., 2017).

5. According the suggestions, the "ratio" and "portion" are revised.

6. According the suggestion, low cases of the "c" are revised.

7. According the suggestion, the contents are revised.

8. According the suggestion, the data description is supplied.

9. HCN vertical resolution is supplied in the data description, same as question 8.

10. Because my result is in summer, that of previous works are in spring next of year, they are not suited to be compare numerical values.

11. This is the cumulative result of the meridional transport.

12. HCN percent is 6

13. According to the suggestion, Figure 1 caption is revised.

14. This is same with Major question 3-4.

15. This is same with Major question 2.

16. According to the suggestion, the horizontal outflow is added.

17. The budget analysis show this result. Because the meridional transport keeps to dilute the role from the tropical troposphere, the role from the mid-latitude of northern hemisphere gradually enhances.

Please also note the supplement to this comment:
https://acp.copernicus.org/preprints/acp-2020-990/acp-2020-990-AC2-supplement.pdf

[Figure]

**Supplement:**

Thank Dr. Diallo for his comments, questions, guidance and suggestions. The replies to his major and minor questions are as follows:

**Replies for the major questions**

1. Equation (5) ignores the time tendency term. If the time tendency term remains, it can be treated as a source or sink term. When the tendency is positive, it is a sink; when the tendency is negative, it is a source. Its relative influence is

$$-\frac{1}{a+b}\frac{1}{\bar{c}}\frac{\partial \bar{c}}{\partial t}$$

According to Figure 3 of Rendel et al. (2010), the HCN difference between June and September is less than 0.02 ppbv, and the HCN average concentration of HCN is more than 0.2 ppbv in the tropical. The air-mass from mid latitude of northern hemisphere can be transported to the tropical within one month, $1 / (a + b)$ is less than $1 / a$, and $1 / a$ is taken as 1 month; the average duration is 4 months (June to September), so the influence of time tendency term is about less than 2.5%.

The chemical process of HCN is depletion role, which is a sink. Its chemical lifetime is about 4 years (Park et al., 2013). The CAM5 model results show that the chemical lifetime of HCN is 2.91 years (about 35 months) at 90 hPa in the tropical. The model top is 1.8 hPa, and vertical levels are 56 layers, the model including troposphere and stratosphere chemistry. Its influence is

$$\frac{1}{a+b}\frac{\bar{Q}}{\bar{c}}$$

which is about 2.9%. Compared with the roles of the transportation from meridional and vertical directions, the influences of the time tendency time and sink term are smaller. So, both are ignored.

[Figure]

**Fig. 3.** Color contours show latitude-time variations of HCN mixing ratio (ppbv) for the lower stratosphere layer, 16 to 23 km, measured by the Aura MLS satellite during September 2004 to November 2009. These MLS data are zonal mean values averaged over individual week periods, as described in (22). Colored crosses indicate HCN for the 16- to 23-km layer derived from the ACE satellite measurements, with each cross indicating an individual profile measurement. Comparison of the MLS and ACE-FTS data are shown in fig. S3.

This figure is from work of Rendel et al. (2010)

2. The contribution of the vertical transport to the local is 1 (=100%), and takes as reference. G is total transportation that includes the contribution to local and meridional transport, and is equal to e / (e-d) ( > 1). The contribution of the vertical transport to meridional transport is d / (e-d), that is equal to G-1. So, G is closed. The mass is conserved.

3-4. There are several ways to define the scope of ASM. Because of the coarse resolution of data, it is difficult to define the range of ASM accurately. However, the upward transportation of the ASM is the key of the ASM, so it is OK to determine the main area of AMS, which includes the upward transport area of the ASM. Therefore, based on wind fields and HCN distributions in the UTLS I identified the 20-40$^{o}$ N region, which includes the ascending region of the AMS. And it can avoid overestimating the impact of the AMS. The observation error of satellite data is less than 10%.

Above 18.5 km, the tropical rising area becomes narrower from 20$^{o}$ S-20$^{o}$ N to 10$^{o}$ S-10$^{o}$ N. First, it can be seen from Figure 1. Second, if 20$^{o}$ S-20$^{o}$ N was taken as the rising region, the distribution of HCN does not satisfy the equations (10) and (12). In addition, plodger et al. (2017) selected 10$^{o}$ S-10$^{o}$ N as the tropical rising region. Therefore, the tropical rising region becomes 10$^{o}$ S-10$^{o}$ N above 18.5 km. The ASM area is still 20-40$^{o}$ N. The case of region 10-20$^{o}$ N needs to be considered. Since the rising speed of this region is 0, the meridional variation is 0 in 10- 20$^{o}$ N ( $\frac{\partial \bar{c}}{\partial y} = 0$). Therefore, HCN in this region is equal to HCN from 20-40$^{o}$ N. The two areas are put together to treat for easy.

5. According the suggestions, the paper is revised.

**Replies for the minor questions**

1. According to the suggestion, The title is revised.
2. According to the suggestion, The "deep stratosphere" is revised.
3. According to the suggestion, the relative contents are revised.
4. I think that there are differences between modeled and observed HCN in the southern hemisphere and tropical middle stratosphere in summer (Park et al., 2013; Ploeger et al., 2017).
5. According the suggestions, the "ratio" and "portion" are revised.
6. According the suggestion, low cases of the "c" are revised.
7. According the suggestion, the contents are revised.
8. According the suggestion, the data description is supplied.
9. HCN vertical resolution is supplied in the data description, same as question 8.
10. Because my result is in summer, that of previous works are in spring next of year, they are not suited to be compare numerical values.
11. This is the cumulative result of the meridional transport.
12. HCN percent is 6% higher than the air-mass because HCN concentration from the ASM is higher than that from the tropical.
13. According to the suggestion, Figure 1 caption is revised.

14. This is same with Major question 3-4.

15. This is same with Major question 2.

16. According to the suggestion, the horizontal outflow is added.

17. The budget analysis show this result. Because the meridional transport keeps to dilute the role from the tropical troposphere, the role from the mid-latitude of northern hemisphere gradually enhances.

---

## Author Comment (AC3) · 1 Mar 2021

First of all, thank the referee for reviewing my manuscript again. The replies to his comments and questions are as follows:

1. Equation (3) is derived from the three-dimensional mass conservation Equation (1). The purpose of comparing Equation (3) with Equation (4) (Andrews et al., 1987) is to further illustrate Equation (3), and is not to explain Equation (4).

2. Under the steady state, Equation (5) is obtained from Equation (3). Equation (5) means that the net input or output (on the left side of the equation) is equal to the sink or source (on the right side of the equation), which indicates that the mass is conserved. There is no assumption that "the transport is proportional to the mean

gradient".

3. a represents the features of meridional transport (unit: 1 / s), b represents the features of vertical transport. F = a /(a + b) means the fraction of meridional transportation in the total transportation. This is consistent with the explanation in the paper. Therefore, the first term on the right side of Equation (10) indicates the contribution of meridional transport, and the second term indicates the contribution of vertical transport.

4. In introduction section, I introduced the work of Rendel et al.(2010). Since HCN is used as a tracer, Figure 1 is the same as Figure 2 of Rendel et al., but the depiction contents are different. Figure 2 is similar to Figure 1a of Rendel et al., but what I illustrate and discuss is different with that of Rendel et al. Their work shows that the ASM is an important transport pathway from the troposphere to the middle and upper stratosphere in summer. My job is to estimate the contribution of air-mass via the ASM into the middle and upper stratosphere.